# ShuffleMixer: An Efficient ConvNet for Image Super-Resolution

**Long Sun, Jinshan Pan**[*]**, Jinhui Tang**[*]
Nanjing University of Science and Technology
{cs.longsun, jspan, jinhuitang}@njust.edu.cn

## Abstract

Lightweight and efficiency are critical drivers for the practical application of image super-resolution (SR) algorithms. We propose a simple and effective approach, ShuffleMixer, for lightweight image super-resolution that explores large convolution and channel split-shuffle operation. In contrast to previous SR models that simply stack multiple small kernel convolutions or complex operators to learn representations, we explore a large kernel ConvNet for mobile-friendly SR design. Specifically, we develop a large depth-wise convolution and two projection layers based on channel splitting and shuffling as the basic component to mix features efficiently. Since the contexts of natural images are strongly locally correlated, using large depth-wise convolutions only is insufficient to reconstruct fine details. To overcome this problem while maintaining the efficiency of the proposed module, we introduce Fused-MBConvs into the proposed network to model the local connectivity of different features. Experimental results demonstrate that the proposed ShuffleMixer is about $3\times$ smaller than the state-of-the-art efficient SR methods, e.g. CARN, in terms of model parameters and FLOPs while achieving competitive performance. The code is available at https://github.com/sunny2109/ShuffleMixer.

## 1 Introduction

Single image super-resolution (SISR) aims to recover a high-resolution image from a low-resolution one. This is a classical problem that has attracted lots of attention recently due to the rapid development of high-definition devices, such as Ultra-High Definition Television, Samsung Galaxy S22 Ultra, iPhone 13 Pro Max, and HUAWEI P50 Pro, and so on. Thus, it is of great interest to develop an efficient and effective method to estimate high-resolution images to be better displayed on these devices.

Recently, convolutional neural network (CNN) based SR models [8, 9, 1, 16, 25, 45] have achieved impressive reconstruction performance. However, these networks hierarchically extract local features, which highly rely on stacking deeper or more complex models to enlarge the receptive fields for performance improvements. As a result, the required computational budget makes these heavy SR models challenging to deploy on resource-constrained mobile devices in practical applications [44].

To alleviate heavy SR models, various methods have been proposed to reduce model complexity or speed up runtime, including efficient operation design [32, 28, 36, 9, 16, 1, 33, 43, 23, 27], neural architecture search [6, 35], knowledge distillation [12, 13], and structural re-parameterization methodology [7, 23, 44]. These methods are mainly based on improved small spatial convolutions or advanced training strategies, and large kernel convolutions are rarely explored. Moreover, they mostly focus on one of the efficiency indicators and do not perform well in real resource-constrained tasks. Thus, the demand for a better trade-off between complexity, latency, and SR quality is imperative.

---

[*]Jinshan Pan and Jinhui Tang are the corresponding authors.

36th Conference on Neural Information Processing Systems (NeurIPS 2022).

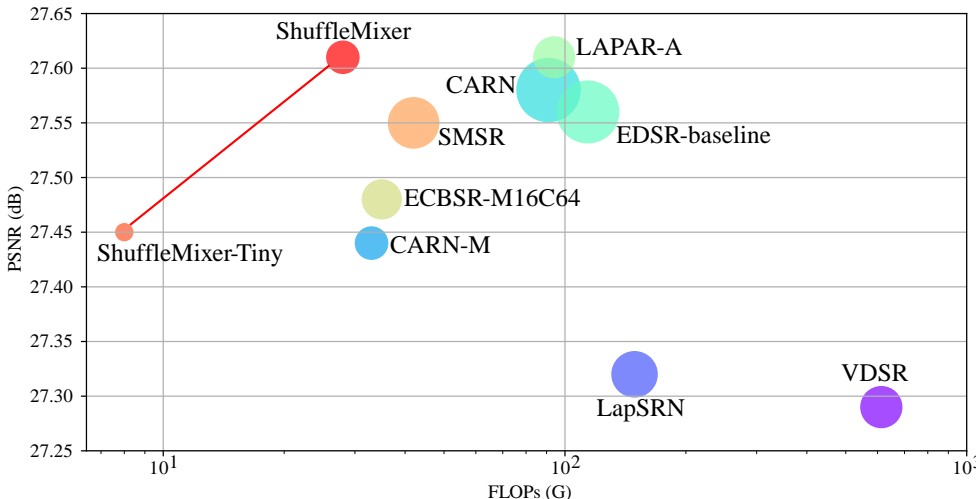

Figure 1: Model complexity and performance comparison between our proposed ShuffleMixer family and other lightweight methods on B100 [2] for $\times 4$ SR. Circle sizes indicate the number of parameters. ShuffleMixer achieves a better trade-off.

A large receptive field involves more feature interactions, which helps reconstruct more refined results in tasks such as super-resolution that require dense per-pixel predictions. Recent visual transformer (ViT)-based approaches [10, 26, 30, 24] employ a multi-head self-attention (MHSA) mechanism to learn long-range feature representations, which leads to their state-of-the-art performance in various vision tasks. However, MHSA is not friendly to enlarging the receptive field of an efficient SR design. Its complexity grows quadratically with the input resolution (the size is usually large and constant during SR training). Regular convolution with large kernels is also a simple but heavyweight approach to obtaining efficient receptive fields. To make large kernel convolutions practical, using depth-wise convolutions with large kernel sizes [27, 39, 7] is an effective alternative. Since depth-wise convolutions share connection weights between spatial locations and remain independent between channels, this property makes it challenging to capture sufficient interactions. It is essential to improve the learning capability of depth-wise convolutions (DW Convs) in lightweight network design.

In this paper, we develop a simple and effective network named ShuffleMixer that introduces large kernel convolutions for lightweight SR design. The core idea is to fuse non-local and local spatial locations within a feature mixing block with fewer parameters and FLOPs. Specifically, we employ depth-wise convolutions with large kernel sizes to aggregate spatial information from a large region. For channel mixing, we introduce channel splitting and shuffling strategies to reduce model parameters and computational costs and improve network capability. We then build an effective shuffle mixer layer based on these two operators. To further improve the learning capability, we embed the Fused-MBConv into the mixer layer to boost local connectivity. Taken together, we find that the ShuffleMixer network with a simple module can obtain state-of-the-art performance. Figure 1 shows that our ShuffleMixer achieves a better trade-off with the least parameters and FLOPs among all existing lightweight SR methods.

The contributions of this paper are summarized as follows: (1) We develop an efficient SR design by exploring a large kernel ConvNet that involves more useful information for image SR. (2) We introduce a channel splitting and shuffling operation to perform feature mixing of the channel projection efficiently. (3) To better explore the local connectivity among cross-group features from the shuffle mixer layer, we utilize Fused-MBConvs in the proposed SR design. We formulate the aforementioned modules into an end-to-end trainable network, which is named as ShuffleMixer. Experimental results show that ShuffleMixer is about $3\times$ smaller than the state-of-the-art methods in terms of model parameters and FLOPs while achieving competitive performance compared to the state-of-the-art methods.

## 2   Related Work

**CNN-based Efficient SR.** CNN-based methods adopt various ways to reduce model complexity. FSRCNN [9] and ESPCN [33] employ post-upsampling layers to significantly reduce the compu-

tational burden from predefined inputs. Namhyuk *et al.* [1] uses group convolution and cascading connection upon a recursive network to save parameters. Hui *et al.* [16] proposes a lightweight information multi-distillation network (IMDN) to aggregate features by applying feature splitting and concatenation operations, and the improved IMDN variants [43, 23] won the AIM2020 and NTIRE2022 Efficient SR challenge. Meanwhile, an increasingly popular approach is to search for a well-constrained architecture as a multi-objective evolution problem [6, 35]. Another branch compresses and accelerates a heavy deep model through knowledge distillation [13, 12]. Note that fewer parameters and FLOPs do not sufficiently mean faster runtime on mobile devices because FLOPs ignore important latency-related factors such as memory access cost (MAC) and degree of parallelism [28, 32]. In this paper, we analyze factors affecting the efficiency of SR models and develop a mobile-friendly SR network.

**Transformer-based SR.** Transformers were initially proposed for language tasks, which stacked the multi-head self-attention and feed-forward MLP layers to learn long-range relations among its inputs. Dosovitskiy *et al.* [10] first applied a vision transformer to image recognition. Since then, ViT-based models have become increasingly applicable to both high-level and low-level vision tasks. For image super-resolution, Chen *et al.* [4] develop a pre-trained image processing transformer (IPT) that directly applies the vanilla ViT to non-overlapped patch embeddings. Liang *et al.* [24] follow Swin Transformer [26] and propose a window-based self-attention model for image restoration tasks and achieve excellent results. Window-based self-attention is much more computationally efficient than global self-attention, but it is still a time-consuming and memory-intensive operation.

**Models with Large Kernels.** AlexNet [20] is a classic large-kernel convolutional neural network model that inspired many subsequent works. Global Convolutional Network [31] uses symmetric, separable large filters to improve semantic segmentation performance. Due to the high computational cost and a large number of parameters, large-size convolutional filters became not popular after VGG-Net [34]. However, large convolution kernels have recently gained attention with the development of efficient convolution techniques and new architectures such as transformers and MLPs. ConvMixer [39] replaces the mixer component of ViTs [26, 10] or MLPs [38] with large kernel depth-wise convolutions. ConvNeXt [27] uses $7 \times 7$ depth-wise kernels to redesign a standard ResNet and achieves comparable results to Transformers. RepLKNet [7] enlarges the convolution kernel to $31 \times 31$ to build a pure CNN model, which obtains better results than Swin Transformer [26] on ImageNet. Unlike these methods that focus on building big models for high-level vision tasks, we explore the possibility of large convolution kernels for lightweight model design in image super-resolution.

# 3   Proposed Method

We aim to develop an efficient large-kernel CNN model for the SISR task. To this end, we introduce key designs to the feature mixing block employed to encode information efficiently. In this section, we first present the overall pipeline of our proposed ShuffleMixer network. Then, we formulate the feature mixing block, which acts as a basic module for building the ShuffleMixer network. Finally, we provide details on the training loss function.

## 3.1   ShuffleMixer architecture

**The overall ShuffleMixer architecture.** Given a low-resolution image $I_{LR} \in \mathbb{R}^{C \times H \times W}$, where $C$, $H \times W$ denote the number of channels and the spatial resolution, respectively. For a color image, the value of $C$ is 3. The proposed ShuffleMixer first extracts feature $Z_0 \in \mathbb{R}^{D \times H \times W}$ by a convolution operation with a filter size of $3 \times 3$ pixels and $D$ channels. Then, we develop a feature mixing block (FMB) consisting of a Fused-MBConv [36] and two shuffle mixer layers, which takes the feature $Z_0$ as input to produce a deeper feature $Z_1 \in \mathbb{R}^{D \times H \times W}$. Next, we utilize an upsampler module with a scale factor $s$ to upscale the spatial resolution of the features generated by a sequence of FMBs. To save parameters of the enlargement module as much as possible, we only use a convolutional layer of size $1 \times 1$ and a pixel shuffling layer [33]. For the $\times 4$ scale factor, we progressively upsample the resolution by repeating the $\times 2$ upsampler two times. Finally, we use a convolutional layer to map the upscaled feature to the residual image $I_R \in \mathbb{R}^{C \times sH \times sW}$, and add it to the upscaled $I_{LR}$ by bilinear interpolation to get the final high-resolution image $I_{SR}$: $I_{SR} = \uparrow^s (I_{LR}) + I_R$, where $\uparrow^s (\cdot)$ denotes the bilinear interpolation with scale factor $s$. In the following, we explain the proposed method in detail.

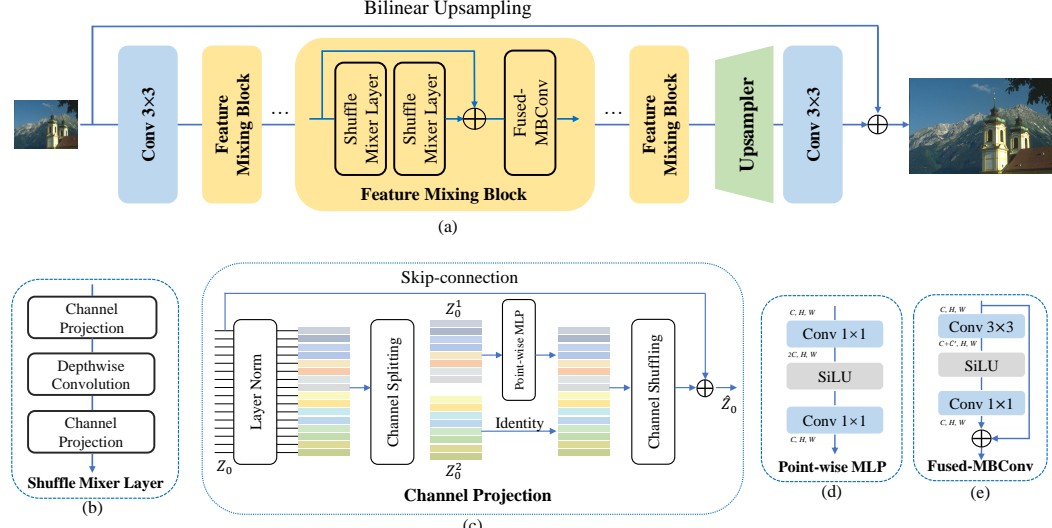

Figure 2: An overview (a) of the proposed ShuffleMixer. The input low-resolution image is first converted to feature space by a convolution layer with a kernel size of $3 \times 3$ pixels. Sequential Feature Mixing Block (FMB) modules are then applied to extract representative features and upsample them for final feature reconstruction. The FMB module contains one (e) Fused-MBConv and two (b) shuffle mixer layers. Each shuffle mixer layer has a large kernel depth-wise convolution and two (c) channel projection submodules. Other components include: channel splitting and shuffling operations, (d) point-wise MLP layers, skip connections, and LayerNorm on the channels.

**The Feature Mixing Block** is developed to explore local and non-local information for better feature representations. To obtain non-local feature interactions effectively, we apply shuffle mixer layers on $Z_0$, as illustrated in Figure 2(a). For each shuffle mixer layer, we employ a large kernel DW Conv to mix features at spatial locations. This operation enjoys large receptive fields with fewer parameters, which can encode more spatial information for reconstructing complete and accurate structures. As we investigated in Table 3, depth-wise convolutions with larger sizes consistently improve SR performance while maintaining computational efficiency.

To mix features at channel locations, we employ point-wise MLPs to perform channel projection. With the help of depth-wise convolution, the computational cost of the shuffle mixer layer is mainly caused by channel projections. We further introduce a channel splitting and shuffling (CSS) strategy [28] to improve the efficiency of this step. Specifically, the input feature $Z_0$ is first to split into $Z_0^1$ and $Z_0^2$; then, a point-wise MLP performs channel mixing on the split feature $Z_0^1$; finally, a channel shuffling operation is employed to enable the exchange of information on the concatenate feature. Therefore, the parameter complexity of the channel projection layer drops from $\Omega(4C^2)$ to $\Omega(C^2)$. This procedure can be formulated as follow:

$$
\begin{aligned}
[Z_0^1, Z_0^2] &= \text{Split}(\text{LayerNorm}(Z_0)) \\
\hat{Z}_0^1 &= W_1(\sigma(W_0(Z_0^1))))) \\
\hat{Z}_0 &= \text{Shuffle}([\hat{Z}_0^1, Z_0^2]) + Z_0
\end{aligned}
\tag{1}
$$

where $\sigma$ is SiLU nonlinearity function [11], $W_0$ and $W_1$ are the point-wise convolutions; $\text{Split}(\cdot)$ and $\text{Shuffle}(\cdot)$ represent the splitting and shuffling of features in the channel dimension. This splitting operation limits representational capability since we exclude the other half of the input tensors from channel interactions. The channel shuffle operation cannot guarantee that all features are processed. Inspired by the MobileNetv2 block [32], we thus repeat the channel projection layer and arrange them before and after the large depth-wise convolution to learn visual representations. From our study, as listed in Table 2, the enhanced mixer layer achieves quite similar results to the ConvMixer block [39] while using fewer parameters and FLOPs.

Since the content of natural images is locally correlated, the stacked Shuffle Mixer Layers do not fully exploit local features. It requires more capacity to model feature representations for better SR performance. Therefore, we embed a few convolutional blocks into the proposed model to enhance local connectivity. Concretely, we evenly add the Fused-MBConv after every two shuffle mixer layers. The original Fused-MBConv contains an expansion convolution of size $3 \times 3$, an SE layer [14] (i.e., the commonly used channel attention), and a reduction convolution of size $1 \times 1$. Using such a Fused-MBConv significantly increases parameters and FLOPs, which motivates us to make some changes to match the computational requirements. We remove the SE layer first, as the SiLU function can be somewhat treated as a gating mechanism. Note that the inference time is much slower as the hidden dimension expands. Instead of expanding the hidden channel rapidly with a large factor (where the default expansion factor is usually set to be 6) of this expansion convolution, we then limit the number of output channels and expand it to $C + C'$ ($C'$ is experimentally set to 16), as shown in Figure 2(d). We also study several operations for this mixing process, and more details are included in Sec. 4.3.

### 3.2 Learning strategy

To constrain the network training, a straightforward way is to ensure that the content of the network output is close to that of the ground truth image:

$$\mathcal{L}_p = \|I_{SR} - I_{GT}\|_1, \tag{2}$$

where $I_{SR}$ and $I_{GT}$ denote the network output and the corresponding ground truth HR image. We note that only using the above pixel-wise loss function does not effectively help high-frequency details estimation [5]. We accordingly employ a frequency constraint to regularize network training, where the final loss function is defined as:

$$\mathcal{L} = \mathcal{L}_p + \lambda\|\mathcal{F}(I_{SR}) - \mathcal{F}(I_{GT})\|_1, \tag{3}$$

where $\mathcal{F}$ denotes the Fast Fourier transform, and $\lambda$ is a weight parameter that is set to be 0.1 empirically.

## 4 Experimental Results

### 4.1 Datasets and implementation

**Datasets.** Following existing methods [22, 24, 23], we train our models on the DF2K dataset, a merged dataset with DIV2K [37] and Flickr2K [25], which contains 3450 (800 + 2650) high-quality images. We adopt standard protocols to generate LR images by bicubic downscaling of reference HR images. During the testing stage, we evaluate our models with the peak signal to noise ratio (PSNR) and the structural similarity index (SSIM) on five publicly available benchmark datasets - Set5 [3], Set14 [42], B100 [2], Urban100 [15] and Manga109 [29]. All PSNR and SSIM results are calculated on the **Y** channel from the YCbCr color space.

**Implementation details.** We train our model in RGB channels and augment the input patches with random horizontal flips and rotations. In each training mini-batch, we randomly crop 64 patches of size $64 \times 64$ from LR images as the input. The proposed model is trained by minimizing L1 loss and the frequency loss [5] with Adam [19] optimizer for 300,000 total iterations. The learning rate is set to a constant $5 \times 10^{-4}$. All experiments are conducted with the PyTorch framework on an Nvidia Tesla V100 GPU.

We provide two models according to the number of feature channels and DW Conv kernel size, and the number of FMB modules is 5. The number of channels and convolution kernel sizes is 64 and $7 \times 7$ pixels for the ShuffleMixer model and 32 and $3 \times 3$ pixels for the ShuffleMixer-Tiny model.

### 4.2 Comparisons with state-of-the-art methods

To evaluate the performance of our approach, we compare the proposed ShuffleMixer with state-of-the-art lightweight algorithms, including SRCNN [8], FSRCNN [9], ESPCN [33], VDSR [18], DRCN [17], LapSRN [21], CARN [1], EDSR-baseline [25], FALSR-A [6], IMDN [16], LAPAR [22], ECBSR [44], and SMSR [40].

Table 1: Comparisons on multiple benchmark datasets for efficient SR networks. All results are calculated on the **Y**-channel. The FLOPs are calculated corresponding to an HR image of size $1280 \times 720$. Best and second-best performance are in remarked red and blue color, respectively. Blanked entries link to results not reported in previous works.

| Scale | Method | Params | FLOPs | Set5 | Set14 | B100 | Urban100 | Manga109 |
|---|---|---|---|---|---|---|---|---|
| ×2 | Bicubic | - | - | 33.66/0.9299 | 30.24/0.8688 | 29.56/0.8431 | 26.88/0.8403 | 30.80/0.9339 |
| | SRCNN [8] | 57K | 53G | 36.66/0.9542 | 32.42/0.9063 | 31.36/0.8879 | 29.50/0.8946 | 35.74/0.9661 |
| | FSRCNN [9] | 12K | 6G | 37.00/0.9558 | 32.63/0.9088 | 31.53/0.8920 | 29.88/0.9020 | 36.67/0.9694 |
| | ESPCN [33] | 21K | 5G | 36.83/0.9564 | 32.40/0.9096 | 31.29/0.8917 | 29.48/0.8975 | - |
| | VDSR [18] | 665K | 613G | 37.53/0.9587 | 33.03/0.9124 | 31.90/0.8960 | 30.76/0.9140 | 37.22/0.9729 |
| | DRCN [17] | 1,774K | 17,974G | 37.63/0.9588 | 33.04/0.9118 | 31.85/0.8942 | 30.75/0.9133 | 37.63/0.9723 |
| | LapSRN [21] | 813K | 30G | 37.52/0.9590 | 33.08/0.9130 | 31.80/0.8950 | 30.41/0.9100 | 37.27/0.9740 |
| | CARN-M [1] | 412K | 91G | 37.53/0.9583 | 33.26/0.9141 | 31.92/0.8960 | 31.23/0.9193 | - |
| | CARN [1] | 1,592K | 223G | 37.76/0.9590 | 33.52/0.9166 | 32.09/0.8978 | 31.92/0.9256 | - |
| | EDSR-baseline [25] | 1,370K | 316G | 37.99/0.9604 | 33.57/0.9175 | 32.16/0.8994 | 31.98/0.9272 | 38.54/0.9769 |
| | FALSR-A [6] | 1021K | 235G | 37.82/0.9595 | 33.55/0.9168 | 32.12/0.8987 | 31.93/0.9256 | - |
| | IMDN [16] | 694K | 161G | 38.00/0.9605 | 33.63/0.9177 | 32.19/0.8996 | 32.17/0.9283 | 38.88/0.9774 |
| | LAPAR-C [22] | 87K | 35G | 37.65/0.9593 | 33.20/0.9141 | 31.95/0.8969 | 31.10/0.9178 | 37.75/0.9752 |
| | LAPAR-A [22] | 548K | 171G | 38.01/0.9605 | 33.62/0.9183 | 32.19/0.8999 | 32.10/0.9283 | 38.67/0.9772 |
| | ECBSR-M16C64 [44] | 596K | 137G | 37.90/0.9615 | 33.34/0.9178 | 32.10/0.9018 | 31.71/0.9250 | - |
| | SMSR [40] | 985K | 132G | 38.00/0.9601 | 33.64/0.9179 | 32.17/0.8990 | 32.19/0.9284 | 38.76/0.9771 |
| | **ShuffleMixer-Tiny (Ours)** | 108K | 25G | 37.85/0.9600 | 33.33/0.9153 | 31.99/0.8972 | 31.22/0.9183 | 38.25/0.9761 |
| | **ShuffleMixer (Ours)** | 394K | 91G | 38.01/0.9606 | 33.63/0.9180 | 32.17/0.8995 | 31.89/0.9257 | 38.83/0.9774 |
| ×3 | Bicubic | - | - | 30.39/0.8682 | 27.55/0.7742 | 27.21/0.7385 | 24.46/0.7349 | 26.95/0.8556 |
| | SRCNN [8] | 57K | 53G | 32.75/0.9090 | 29.28/0.8209 | 28.41/0.7863 | 26.24/0.7989 | 30.59/0.9107 |
| | FSRCNN [9] | 12K | 5G | 33.16/0.9140 | 29.43/0.8242 | 28.53/0.7910 | 26.43/0.8080 | 30.98/0.9212 |
| | VDSR [18] | 665K | 613G | 33.66/0.9213 | 29.77/0.8314 | 28.82/0.7976 | 27.14/0.8279 | 32.01/0.9310 |
| | DRCN [17] | 1,774K | 17,974G | 33.82/0.9226 | 29.76/0.8311 | 28.80/0.7963 | 27.15/0.8276 | 32.31/0.9328 |
| | CARN-M [1] | 412K | 46G | 33.99/0.9236 | 30.08/0.8367 | 28.91/0.8000 | 27.55/0.8385 | - |
| | CARN [1] | 1,592K | 119G | 34.29/0.9255 | 30.29/0.8407 | 29.06/0.8034 | 28.06/0.8493 | - |
| | EDSR-baseline [25] | 1,555K | 160G | 34.37/0.9270 | 30.28/0.8417 | 29.09/0.8052 | 28.15/0.8527 | 33.45/0.9439 |
| | IMDN [16] | 703K | 72G | 34.36/0.9270 | 30.32/0.8417 | 29.09/0.8046 | 28.17/0.8519 | 33.61/0.9445 |
| | LAPAR-C [22] | 99K | 28G | 33.91/0.9235 | 30.02/0.8358 | 28.90/0.7998 | 27.42/0.8355 | 32.54/0.9373 |
| | LAPAR-A [22] | 594K | 114G | 34.36/0.9267 | 30.34/0.8421 | 29.11/0.8054 | 28.15/0.8523 | 33.51/0.9441 |
| | SMSR [40] | 993K | 68G | 34.40/0.9270 | 30.33/0.8412 | 29.10/0.8050 | 28.25/0.8536 | 33.68/0.9445 |
| | **ShuffleMixer-Tiny (Ours)** | 114K | 12G | 34.07/0.9250 | 30.14/0.8382 | 28.94/0.8009 | 27.54/0.8373 | 33.03/0.9400 |
| | **ShuffleMixer (Ours)** | 415K | 43G | 34.40/0.9272 | 30.37/0.8423 | 29.12/0.8051 | 28.08/0.8498 | 33.69/0.9448 |
| ×4 | Bicubic | - | - | 28.42/0.8104 | 26.00/0.7027 | 25.96/0.6675 | 23.14/0.6577 | 24.89/0.7866 |
| | SRCNN [8] | 57K | 53G | 30.48/0.8628 | 27.49/0.7503 | 26.90/0.7101 | 24.52/0.7221 | 27.66/0.8505 |
| | FSRCNN [9] | 12K | 5G | 30.71/0.8657 | 27.59/0.7535 | 26.98/0.7150 | 24.62/0.7280 | 27.90/0.8517 |
| | ESPCN [33] | 25K | 1G | 30.52/0.8697 | 27.42/0.7606 | 26.87/0.7216 | 24.39/0.7241 | - |
| | VDSR [18] | 665K | 613G | 31.35/0.8838 | 28.01/0.7674 | 27.29/0.7251 | 25.18/0.7524 | 28.83/0.8809 |
| | DRCN [17] | 1,774K | 17,974G | 31.53/0.8854 | 28.02/0.7670 | 27.23/0.7233 | 25 .14/0.7510 | 28.98/0.8816 |
| | LapSRN [21] | 813K | 149G | 31.54/0.8850 | 28.19/0.7720 | 27.32/0.7280 | 25.21/0.7560 | 29.09/0.8845 |
| | CARN-M [1] | 412K | 33G | 31.92/0.8903 | 28.42/0.7762 | 27.44/0.7304 | 25.62/0.7694 | - |
| | CARN [1] | 1,592K | 91G | 32.13/0.8937 | 28.60/0.7806 | 27.58/0.7349 | 26.07/0.7837 | - |
| | EDSR-baseline [25] | 1,518K | 114G | 32.09/0.8938 | 28.58/0.7813 | 27.57/0.7357 | 26.04/0.7849 | 30.35/0.9067 |
| | IMDN [16] | 715K | 41G | 32.21/0.8948 | 28.58/0.7811 | 27.56/0.7353 | 26.04/0.7838 | 30.45/0.9075 |
| | LAPAR-C [22] | 115K | 25G | 31.72/0.8884 | 28.31/0.7740 | 27.40/0.7292 | 25.49/0.7651 | 29.50/0.8951 |
| | LAPAR-A [22] | 659K | 94G | 32.15/0.8944 | 28.61/0.7818 | 27.61/0.7366 | 26.14/0.7871 | 30.42/0.9074 |
| | ECBSR-M16C64 [44] | 603K | 35G | 31.92/0.8946 | 28.34/0.7817 | 27.48/0.7393 | 25.81/0.7773 | - |
| | SMSR [40] | 1006K | 42G | 32.12/0.8932 | 28.55/0.7808 | 27.55/0.7351 | 26.11/0.7868 | 30.54/0.9085 |
| | **ShuffleMixer-Tiny (Ours)** | 113K | 8G | 31.88/0.8912 | 28.46/0.7779 | 27.45/0.7313 | 25.66/0.7690 | 29.96/0.9006 |
| | **ShuffleMixer (Ours)** | 411K | 28G | 32.21/0.8953 | 28.66/0.7827 | 27.61/0.7366 | 26.08/0.7835 | 30.65/0.9093 |

Table 1 shows quantitative comparisons on benchmark datasets for the upscaling factors of ×2, ×3, and ×4. In addition to PSNR/SSIM metrics, we list the number of parameters and FLOPs. The number of FLOPs is tested under a setting of super-resolving an image to $1280 \times 720$ pixels. In Figure 1, we compare FLOPs and the number of parameters on the ×4 B100 dataset. Here, our ShuffleMixer model obtains competitive results with even fewer parameters and FLOPs. Especially, ShuffleMixer has a similar number of parameters to CARN-M, but our model outperforms it by a large margin on all benchmark datasets. Even with only 113K parameters, ShuffleMixer-Tiny performs better than many existing methods. With regard to the scale factor ×2 and ×3, the proposed ShuffleMixer family can achieve similar performance.

Although IMDN [16], LAPAR-A [22] and SMSR [40] obtain comparable PSNR/SSIM performance, ShuffleMixer requires only a relatively small amount of model complexity. Meanwhile, we compare

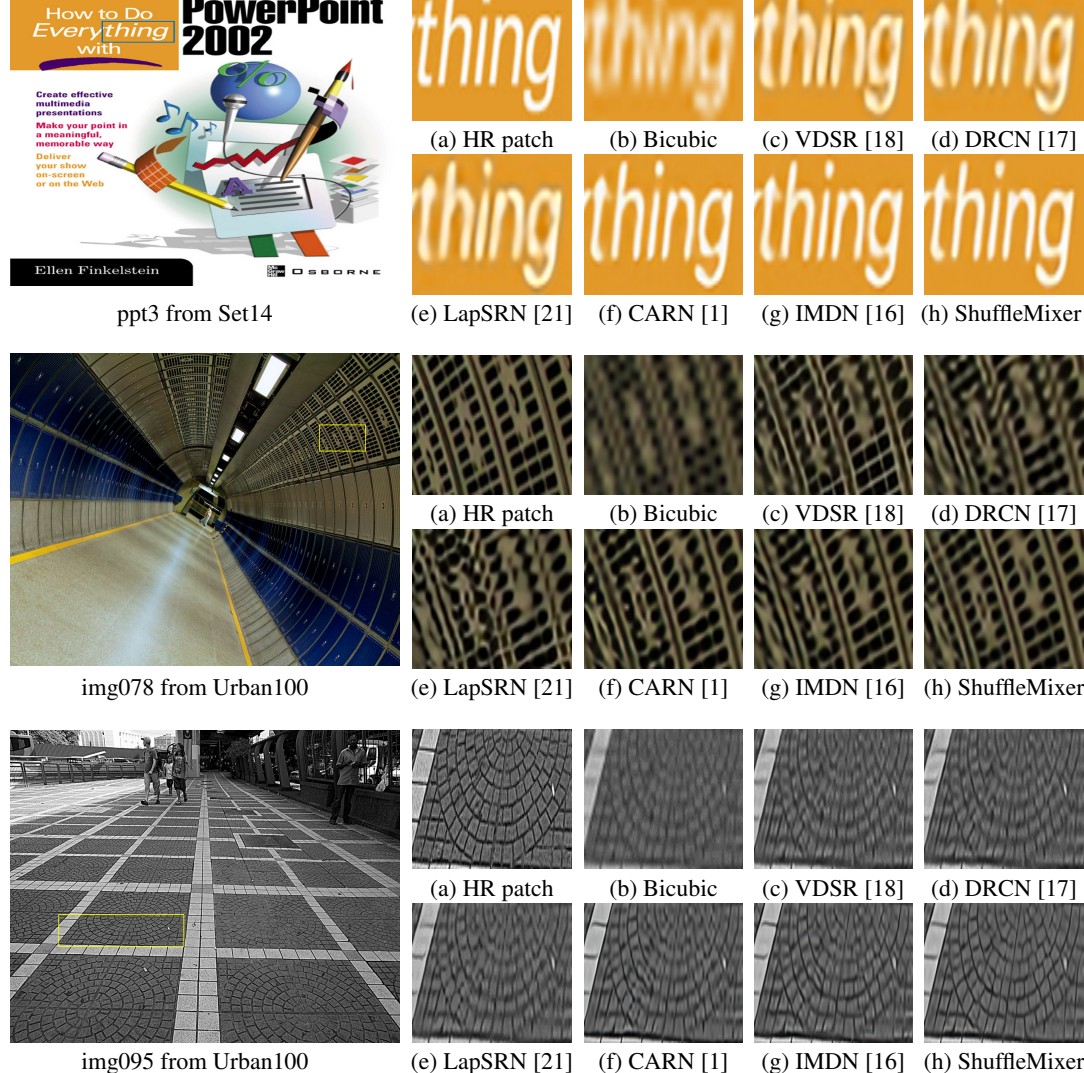

Figure 3: Visual comparisons for ×4 SR on Set14 and Urban100 datasets. The proposed algorithm recovers the image with clearer structures.

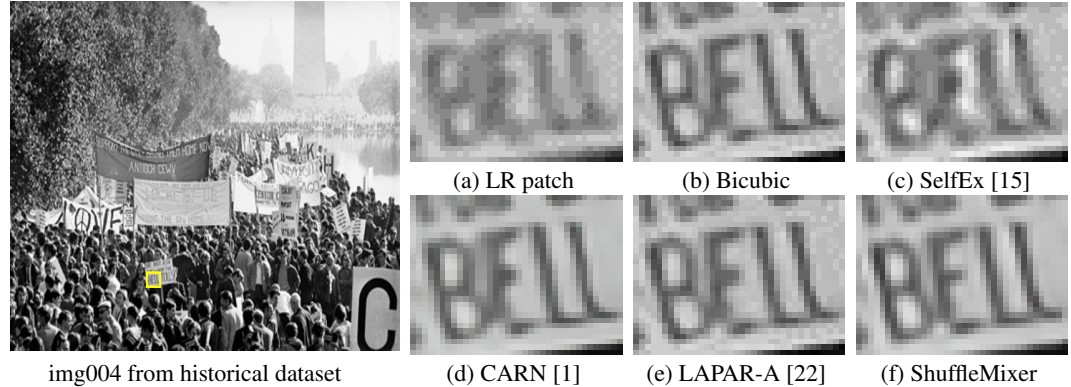

Figure 4: Visual comparisons for ×3 SR on historical dataset [21]. Compared with the results in (b)–(e), the super-resolved image (f) generated by our approach is much clearer with fewer artifacts.

Table 2: Ablation studies of the shuffler mixer layer and the feature mixing block on $\times 4$ DIV2K validation set[37]. The FLOPs are calculated by fvcore on an LR image with a resolution of $256 \times 256$ pixels (FLOPs in Table 3 and Table 4 are also measured with this setting).

| | | (a) Shuffle Mixer Layer | | (b) Feature Mixing Block | | | | |
| | Baseline | CSS | CDC | Conv | S-Conv | C-Conv | S-ResBlock | S-FMBConv |
|---|---|---|---|---|---|---|---|---|
| Params(K) | 55.9 | 24.7 | 35.5 | 81.7 | 81.7 | 128 | 128 | 113 |
| FLOPs(G) | 5.2 | 3.2 | 3.8 | 6.9 | 6.9 | 9.9 | 9.9 | 8.9 |
| PSNR(dB) | 29.96 | 29.83 | 29.99 | 30.12 | 30.16 | 30.20 | 30.24 | 30.21 |
| SSIM | 0.8288 | 0.8231 | 0.8259 | 0.8299 | 0.8305 | 0.8316 | 0.8327 | 0.8321 |

Table 3: Experimental results for different kernel size settings. We test PSNR/SSIM results on the $\times 4$ DIV2K validation set, and the kernel size ranges from $3 \times 3$ to $21 \times 21$ pixels.

| Kernel Size | PSNR(dB)/SSIM | Params(K) | FLOPs(G) |
|---|---|---|---|
| $3 \times 3$ | 30.21/0.8321 | 113 | 8.9 |
| $5 \times 5$ | 30.24/0.8326 | 118 | 9.2 |
| $7 \times 7$ | 30.28/0.8342 | 125 | 9.7 |
| $9 \times 9$ | 30.29/0.8339 | 136 | 10.4 |
| $11 \times 11$ | 30.28/0.8339 | 148 | 11.2 |
| $13 \times 13$ | 30.29/0.8337 | 164 | 12.2 |
| $15 \times 15$ | 30.28/0.8336 | 182 | 13.4 |
| $21 \times 21$ | 30.29/0.8339 | 251 | 17.9 |

Table 4: Effect of increasing the depth (**#B**) or width (**#C**) of the proposed tiny model. All the methods are evaluated on $\times 4$ DIV2K validation set.

| Model | #B | #C | Kernel Size | PSNR(dB)/SSIM | Params(K) | FLOPs(G) |
|---|---|---|---|---|---|---|
| ShuffleMixer-Tiny | 5 | 32 | $3 \times 3$ | 30.21/0.8321 | 113 | 8.9 |
| A | 5 | 32 | $5 \times 5$ | 30.24/0.8326 | 118 | 9.2 |
| B | 5 | 32 | $7 \times 7$ | 30.28/0.8342 | 125 | 9.7 |
| C | 5 | 36 | $3 \times 3$ | 30.25/0.8329 | 138 | 10.8 |
| D | 5 | 40 | $3 \times 3$ | 30.28/0.8339 | 166 | 13.0 |
| E | 6 | 32 | $3 \times 3$ | 30.25/0.8322 | 133 | 10.2 |
| F | 8 | 32 | $3 \times 3$ | 30.30/0.8342 | 174 | 12.8 |

the GPU run time with fast and lightweight models on $\times 4$ SR: CARN [1], CARN-M [1] and LAPAR-A [22], and the proposed method has a faster inference speed. Our ShuffleMixer-Tiny and ShuffleMixer reconstruct an HR image with a resolution of $1280 \times 720$ pixels with 0.016s and 0.021s, respectively. As a comparison, the runtimes are 0.017s, 0.019s, and 0.031s for CARN-M, CARN, and LAPAR-A. Note that Pytorch has poor support for large-kernel depth-wise convolution; employing optimized depth-wise convolutions can further accelerate the inference time of our method, as suggested in [7]. All these results demonstrate the effectiveness of our method.

Figure 3 presents visual comparisons on Set14 and Urban100 datasets for a $\times 4$ scale. The qualitative comparison results demonstrate that our proposed methods can produce more visually pleasing results. The structures and details are better recovered.

We further evaluate our approach on real low-quality images. One example from the historical dataset [21] is shown in Figure 4. The results by [15, 22] show visible artifacts. Our method and CARN [1] generate smooth details, but our results have a clearer structure.

## 4.3 Analysis and discussions

The core idea of ShuffleMixer lies in the shuffle mixer layer, feature mixing block, and large kernel convolution. In this subsection, we evaluate them on the proposed tiny model and train them on the $\times 4$ DIV2K dataset [37].

**Effectiveness of the shuffle mixer layer.** To verify the efficiency of the shuffle mixer, we use 10 ConvMixer [39] blocks to build a baseline model. Unlike the original ConvMixer module, we replace BatchNorm with LayerNorm and apply it only before the point-wise MLP layer because BatchNorm tends to bring artifacts in the generated results [25, 41]. The kernel of depth-wise convolution is set to 3, and the number of channels is 32. When applying the channel splitting and shuffling (CSS) strategy, the number of parameters is reduced from 55.9K to 24.7K, and the performance is also 0.13dB lower than the baseline. This result reflects that the split operation limits the representation capability of the channel projection layer. To compensate for the lack of PSNR, we repeat the CSS-based projection layer to enable more cross-group feature mixing (denoted by CDC). Table 2(a) shows a quantitative comparison where we find that CDC achieves similar performance to the baseline model while reducing parameters from 55.9K to 35.5K and FLOPs from 5.2G to 3.8G.

**Effectiveness of the feature mixing block.** To validate the effectiveness of the proposed feature mixing block, we take the CDC model as a baseline and embed a convolution layer with a kernel size of $3 \times 3$ pixels after each two shuffle mixer layers. This conduct brings a gain of 0.13dB over

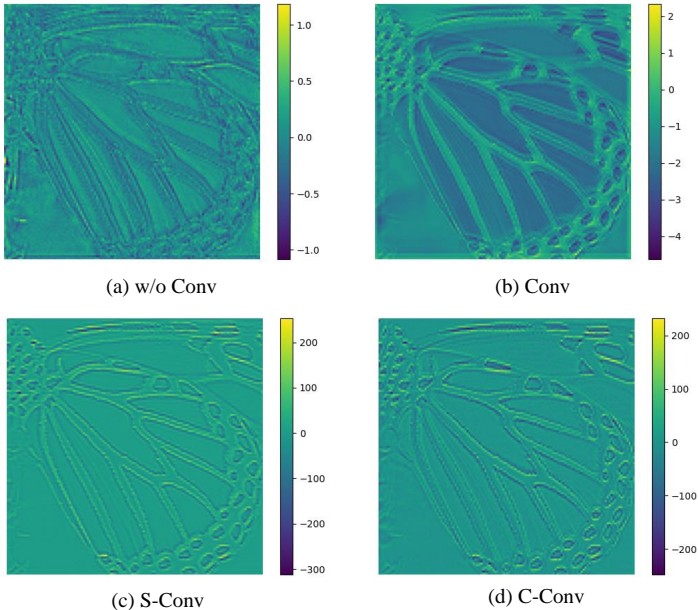

(a) w/o Conv                (b) Conv

(c) S-Conv               (d) C-Conv

Figure 5: Visualization of learned feature maps before the upsampler module. We show the average features over the channel dimension.

the baseline. We then investigate feature element-wise summation or concatenation to analyse the effect of feature fusion manners. The input features are aggregated by the aforementioned ways and refined by a convolutional layer with a kernel size of $3 \times 3$, which we denote as S-Conv and C-Conv, respectively. Table 2(b) shows that they all improve over the baseline; C-Conv achieves better PSNR performance while having more computational cost. Figure 5 exhibits the average feature map of the channel axis before the upsampler module, which illustrates that enhancing local connectivity helps grab more delicate high-frequency contents. Based on the S-Conv, we replace the convolution layer with basic residual blocks (S-ResBlock) and Fused-MBConv (S-FMBConv). Table 2(b) shows that S-FMBConv obtains a balanced trade-off between model complexity and SR performance. Thus, we choose S-FMBConv to strengthen the local interactions within features in this paper.

**Effectiveness of large depth-wise convolution.** To demonstrate the effect of a large kernel, we use different kernel sizes ranging from $3 \times 3$ to $21 \times 21$ pixels and test their performance separately. Table 3 shows that using a larger kernel size will improve the performance. In particular, the PSNR value of the method using a kernel size of $7 \times 7$ pixels is 0.07dB higher than that of using $3 \times 3$ kernel size while only increasing 12K parameters and 0.8G FLOPs. In addition, we note that the performance gains are minor when the kernel size is larger than $7 \times 7$ pixels. Thus, the kernel size is set to be $7 \times 7$ pixels as a trade-off between accuracy and model complexity in this paper.

Moreover, we investigate the effect of increasing the depth (**#B**) or width (**#C**) of the proposed tiny model on the reconstruction performance. Table 4 shows that increasing the depth or width of the model with a small kernel size does not yield as good a performance gain as enlarging the kernel size. These results also demonstrate the effectiveness of employing large depth-wise convolution.

## 5 Conclusion

In this paper, we propose a lightweight deep model to solve the image super-resolution problem. The proposed deep model, i.e., ShuffleMixer, contains a shuffler mixer layer with a larger effective receptive field to extract non-local feature representations efficiently. We also introduce the Fused-MBConv to modulate the local connectivity of features generated by the shuffler mixer layer, which is critical for improving SR performance. We both qualitatively and quantitatively evaluate the proposed ShuffleMixer on commonly used benchmarks. Experimental results demonstrate that the proposed ShuffleMixer is much more efficient while achieving competitive performance than the state-of-the-art methods.

**Acknowledgments.** This work has been supported in part by the National Key R&D Program of China (No. 2018AAA0102001), the National Natural Science Foundation of China (Nos. 61922043, 61872421, 61925204), and the Fundamental Research Funds for the Central Universities (No. 30920041109).

## Broader Impact

This paper is an exploratory work on lightweight and efficient image super-resolution using a large-kernel ConvNet. This approach can be deployed in some resource-constrained environments to improve image quality, such as processing pictures taken by smartphones and reducing bandwidth during video calls or meetings. However, super-resolution technology has also brought some negative effects, such as criminals using this technology to enhance people's facial or body features, thereby allowing identity information to leak. It is worth noting that the positive social impact of image super-resolution far outweighs the potential problems. We call on people to use this technology and its derivative applications without harming the personal interests of the public.

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
