# ShuffleMixer: An Efficient ConvNet for Image Super-Resolution
# – Supplemental Material –

**Long Sun, Jinshan Pan**[*]**, Jinhui Tang**[*]
Nanjing University of Science and Technology
{cs.longsun, jspan, jinhuitang}@njust.edu.cn

## Overview

In this supplemental material, we first provide additional ablation experiments on the activation function, the normalization layer and the introduced frequency loss in Sec. 1. We then provide more visual comparison results in Sec. 2 and further explore applying ShuffleMixer to other image restoration tasks in Sec. 3. In Sec. 4, we make some comments on the results of the Urban100 dataset.

## 1 More experimental results

**Replacing SiLU with ReLU.** ReLU activation is commonly used in CNN-based SR models due to its simplicity and efficiency. SiLU, a smoother variant of ReLU, is increasingly popular in recent ViTs and advanced CNN architectures. According to its definition, SiLU has similar effects to gating mechanisms. The results shown in Table 1 indicate that using SiLU in the proposed model produces better PSNR/SSIM results on public benchmarks, which suggests that SiLU is a viable alternative to ReLU.

**Effect of LayerNorm.** Normalisation is not usually employed in SR tasks, mainly because Batch-Norm tends to introduce artifacts. LayerNorm is an alternative to BatchNorm to avoid inaccurate estimation of statistics on small batch sizes. It is also widely used in recent ViT models with promising results. Following this setting, we also use LayerNorm in our proposed approach and observe that applying LayerNorm leads to better training stability and performance, as presented in Table 1.

**Effectiveness of the FFT Loss.** To enhance the high-frequency details of output images, we introduce a frequency constraint to the SR results via FFT operation. Table 1 shows that adding this loss function leads to noticeable performance improvements on public test datasets.

## 2 More visual comparison results

In this section, we present additional visual comparisons with state-of-the-art methods [5, 1, 4, 6, 3] on Set14 and Urban100 datasets. Figure 2 shows that our method generates more accurate structures than other comparison methods.

We also show the results of our ShuffleMixer model only trained on DIV2K. Table 2 demonstrates that ShuffleMixer still achieves competitive performance among lightweight SISR methods.

## 3 Applications of the proposed ShuffleMixer

Our ShuffleMixer can be applied to other restoration tasks. In this section, we show the proposed ShuffleMixer for color image denoising.

---

[*]Jinshan Pan and Jinhui Tang are the corresponding authors.

36th Conference on Neural Information Processing Systems (NeurIPS 2022).

Table 1: Ablation experiments for components of the Shuffle Mixer Layer and the frequency loss function. We evaluate them respectively on the developed tiny model and train them on the ×4 DIV2K dataset. "A → B" is to replace A with B. "None" means to remove the operation.

| Ablation | Set5 | Set14 | B100 | Urban100 | Manga109 | DIV2K_val100 |
|---|---|---|---|---|---|---|
| **Baseline** | **31.88/0.8908** | **28.43/0.7772** | **27.44/0.7311** | **25.62/0.7687** | **29.95/0.9002** | **30.21/0.8321** |
| SiLU → ReLU | 31.79/0.8897 | 28.39/0.7763 | 27.41/0.7299 | 25.56/0.7653 | 29.78/0.8978 | 30.16/0.8305 |
| LayerNorm → None | 31.84/0.8900 | 28.41/0.7770 | 27.43/0.7308 | 25.60/0.7667 | 29.82/0.8988 | 30.19/0.8313 |
| FFT Loss → None | 31.79/0.8897 | 28.37/0.7764 | 27.41/0.7305 | 25.57/0.7680 | 29.74/0.8989 | 30.17/0.8316 |

Table 2: PSNR(dB) results of ShuffleMixer. red/blue represents the proposed model trained on DIV2K/DF2K.

| Method | Scale | Params | FLOPs | Set5 | Set14 | B100 | Urban100 | Manga109 |
|---|---|---|---|---|---|---|---|---|
| ShuffleMixer | ×2 | 394K | 91G | 37.99/38.01 | 33.55/33.63 | 32.14/32.17 | 31.85/31.89 | 38.53/38.83 |
| | ×3 | 415K | 43G | 34.39/34.40 | 30.35/30.37 | 29.08/29.12 | 28.03/28.08 | 33.42/33.69 |
| | ×4 | 411K | 28G | 32.13/32.21 | 28.62/28.66 | 27.61/27.61 | 26.10/26.08 | 30.47/30.65 |
| ShuffleMixer-Tiny | ×2 | 108K | 25G | 37.79/37.85 | 33.31/33.33 | 31.99/31.99 | 31.26/31.22 | 37.88/38.25 |
| | ×3 | 114K | 12G | 34.02/34.07 | 30.12/30.14 | 28.94/28.94 | 27.57/27.54 | 33.88/33.03 |
| | ×4 | 113K | 8G | 31.88/31.88 | 28.43/28.46 | 27.44/27.45 | 25.62/25.66 | 29.95/29.96 |

Following [8, 10], we synthesize the noisy images from the DIV2K dataset by adding AWGN of the different noise levels. Besides, we removed the ShuffleMixer's upsampler module and optimized the model parameters in the same way as in the SISR task. We evaluate the denoised results on three benchmark datasets: CBSD68, Kodak24, and Urban100.

Table 4 shows that our ShuffleMixer can be easily extended to other low-level tasks and achieve competitive performances. Figure 3 exhibits that the proposed ShuffleMixer effectively removes the heavy noise, and the details of all images are better preserved in our results.

# 4   Some notes on the Urban100 dataset

Our goal in this section is to make some notes on the results of the Urban100 dataset. As shown in Tab.1 of the main paper, our ShuffleMixer obtains poor PSNR performance on the Urban100 dataset compared to other state-of-the-art methods in terms of ×2 and ×3 super-resolution. In particular, our method is 0.28dB and 0.09dB lower than IMDN in terms of PSNR in scale factors of ×2 and ×3, respectively. Nevertheless, we made a visual comparison and found no perceivable difference in perceptual quality. Thus, we reevaluate these results using two commonly-used perceptual metrics: NIQE and LPIPS. Table 3 lists the quantitative comparison of perception-oriented SR results, and the proposed ShuffleMixer achieves quite similar performance to IMDN in terms of NIQE and LPIPS.

*Why does ShuffleMixer perform poorly on PSNR?* This result may be caused by the luminance differences in some local areas. Since PSNR measures pixel-wise difference rather than overall structure, minor differences in luminance (**Y**-channel) can also make significant PSNR differences. To verify this cause, we select several images with significant PSNR differences (over 0.8dB) from the IMDN model for analysis. Taking Figure 1 as an example, we pick two sets of image patches from the selected image that are spatially neighboring and duplicate in most areas and test their PSNR values separately. On Patch A/C, our ShuffleMixer differs significantly from the IMDN results but is very similar on Patch B/D. This experimental result supports our conclusion.

Table 3: Quantitative comparison results on Urban100 datatset.

| Method | Scale | Params | FLOPs | PSNR↑ | NIQE↓ | LPIPS↓ |
|---|---|---|---|---|---|---|
| IMDN [3] | ×2 | 694K | 161G | 32.17 | 4.59 | 0.1132 |
| | ×3 | 703K | 72G | 28.17 | 5.21 | 0.2136 |
| | ×4 | 715K | 41G | 26.04 | 5.69 | 0.2879 |
| ShuffleMixer | ×2 | 394K | 91G | 31.89 | 4.66 | 0.1127 |
| | ×3 | 415K | 43G | 28.08 | 5.32 | 0.2106 |
| | ×4 | 411K | 28G | 26.08 | 5.78 | 0.2859 |

Table 4: PSNR(dB) comparisons with state-of-the-art methods for color image denoising on benchmark datasets.

| Datasets | $\sigma$ | BM3D [2] | DnCNN [8] | IRCNN [9] | FFDNet [10] | DSNet [7] | ShuffleMixer |
|---|---|---|---|---|---|---|---|
| CBSD68 | 15 | 33.52 | 33.90 | 33.86 | 33.87 | 33.91 | 34.05 |
| | 25 | 30.71 | 31.24 | 31.16 | 31.21 | 31.28 | 31.40 |
| | 50 | 27.38 | 27.95 | 27.86 | 27.96 | 28.05 | 28.15 |
| Kodak24 | 15 | 34.28 | 34.60 | 34.69 | 34.63 | 34.63 | 34.87 |
| | 25 | 32.15 | 32.14 | 32.18 | 32.13 | 32.16 | 32.41 |
| | 50 | 28.46 | 28.95 | 28.93 | 28.98 | 29.05 | 29.25 |
| Urban100 | 15 | 33.93 | 32.98 | 33.78 | 33.83 | - | 34.32 |
| | 25 | 31.36 | 30.81 | 31.20 | 31.40 | - | 31.93 |
| | 50 | 27.93 | 27.59 | 27.70 | 28.05 | - | 28.59 |

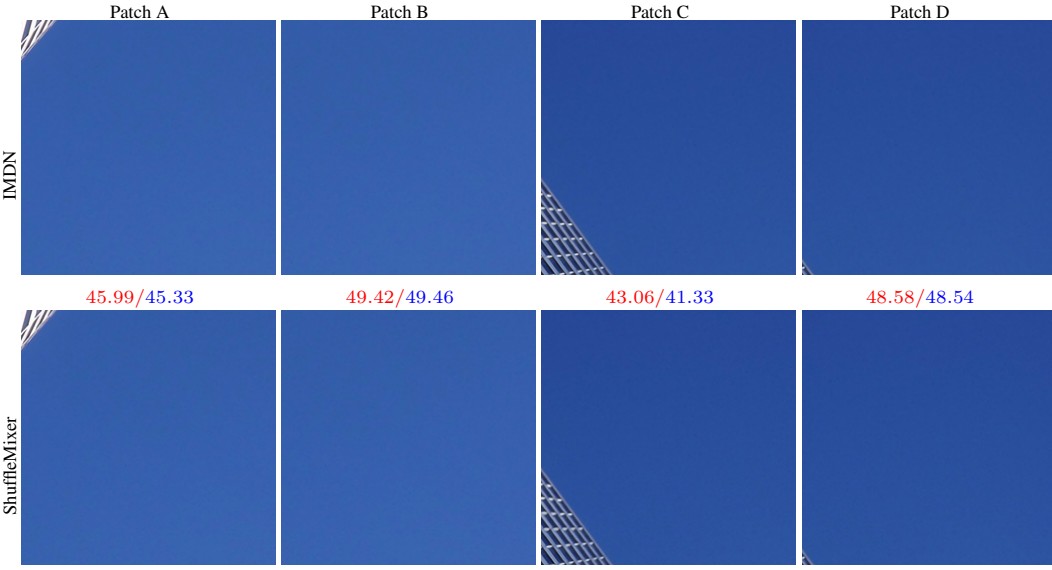

Figure 1: Patch-based visual comparison between IMDN and our ShuffleMixer on the Urban100 dataset. red/blue represents the PSNR results of IMDN/ShuffleMixer.

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

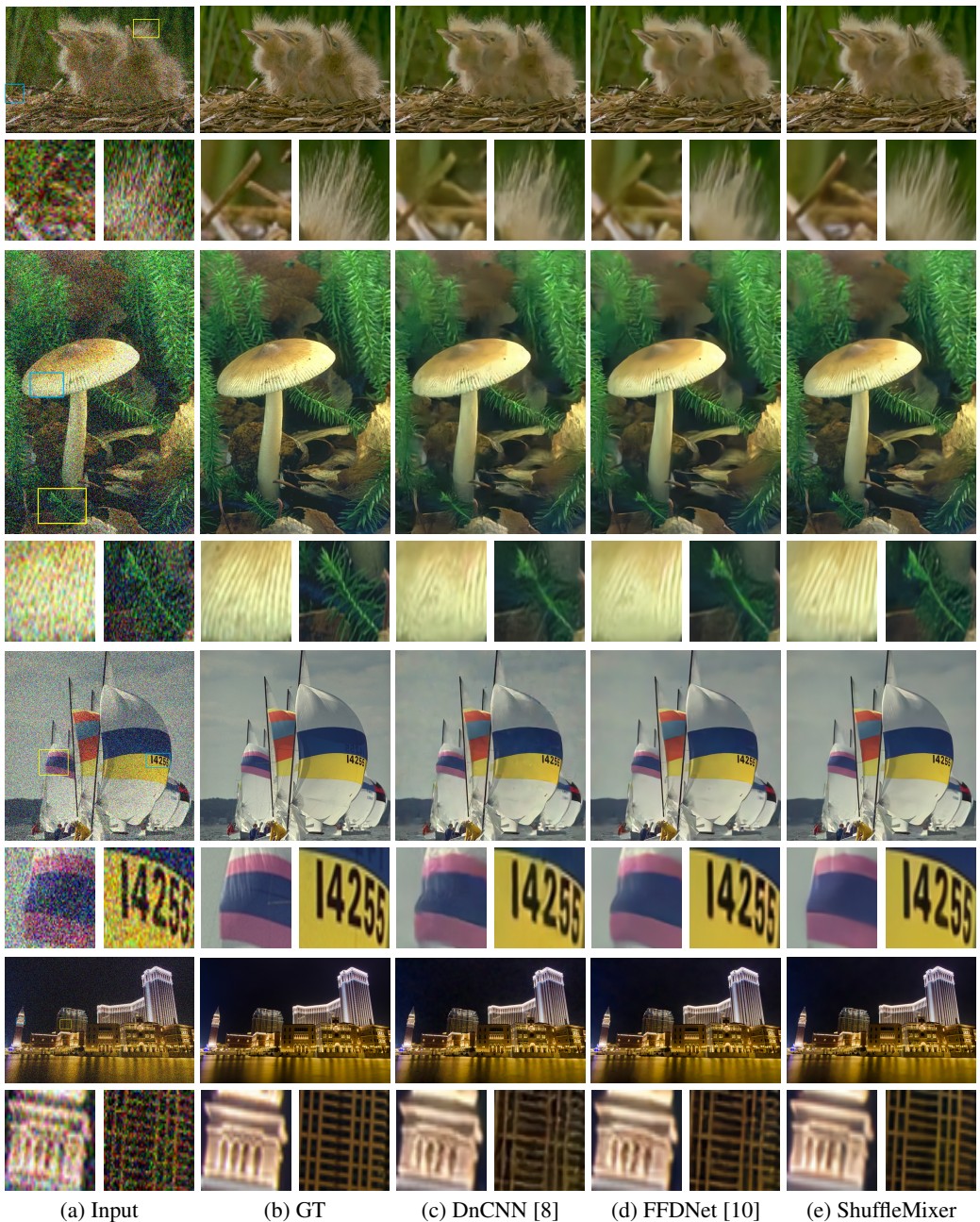

| (a) Input | (b) GT | (c) DnCNN [8] | (d) FFDNet [10] | (e) ShuffleMixer |

Figure 3: Image denoising examples on benchmark datasets. The standard deviation of the noise is set to 50. State-of-the-art methods generate denoised images with over-smoothed results. In contrast, our method preserves better structures and details.

[8] Kai Zhang, Wangmeng Zuo, Yunjin Chen, Deyu Meng, and Lei Zhang. Beyond a Gaussian denoiser: Residual learning of deep CNN for image denoising. *TIP*, 26(7):3142–3155, 2017.

[9] Kai Zhang, Wangmeng Zuo, Shuhang Gu, and Lei Zhang. Learning deep cnn denoiser prior for image restoration. In *CVPR*, 2017.

[10] Kai Zhang, Wangmeng Zuo, and Lei Zhang. FFDNet: Toward a fast and flexible solution for CNN-based image denoising. *TIP*, 27(9):4608–4622, 2018.