# OpenReview forum: "ShuffleMixer: An Efficient ConvNet for Image Super-Resolution"
_NeurIPS.cc/2022/Conference — NeurIPS 2022 Accept_

### Official Review · Reviewer_YLsE · 2022-06-27

**Rating:** 3
**Confidence:** 5
**Soundness:** 3 good
**Presentation:** 3 good
**Contribution:** 1 poor

**Summary:**

This paper proposes an efficient SR network based on depthwise convolutions with large kernel sizes and channel shuffling. The proposed network achieves better performance while with fewer parameters and multi-adds.

**Questions:**

How about the comparisons between speed?  Large kernel size may heavily slow the whole network, which is also unfriendly to terminal devices.

**Ethics Review Area:**

["I don’t know"]

**Limitations:**

The novelty of this paper may be limited and the analysis may be inadequate. See the weakness and questions above.

**Strengths And Weaknesses:**

Strengths:
This paper is well-written and easy to follow. The proposed network can achieve better performance while with fewer parameters and multi-adds.

Weakness:
1. The novelty of this paper may be limited. This paper uses depthwise convolution and channel shuffling to reduce the number of parameters and multi-adds, while these techniques have already been studied by many other methods.

2. The authors argue that a larger receptive field is better for SR. While there are no experiments to analyze the difference between small and large kernels. For example, the authors can perform an ablation study by replacing the large kernels in the proposed network with small kernels and extendinging the depth to keep the model size.

---

> ### Author Response · Authors · 2022-08-02
> **Response to Reviewer YLsE**
>
> We thank the reviewer for the comments and answer the questions below.
>
> **Q1: [Novelty]. This paper uses depthwise convolution and channel shuffling to reduce the number of parameters and multi-adds, while these techniques have already been studied by many other methods.**
>
> We emphasize the novelty and contributions as follows.
>
> Lightweight and efficiency are key drivers for practical applications of image SR algorithms. In contrast to existing methods that directly stack multiple depthwise convolutions (DW Convs) or channel shuffling operations, we develop a lightweight and efficient SR model based on a ViT-like architecture. To achieve non-local feature interactions for better performance, we develop a large kernel convolution based on DW Conv, which allows spatial interaction at a lightweight cost. In addition,we develop two channel shuffling operations to reduce the computational cost of the channel projection module and maintain performance. Experimental results show that the proposed method performs favorably against SOTA methods in terms of model parameters and FLOPs.
>
> The use of the DW Conv is to explore the feasibility of applying large kernel convolution to image SR, whereas the previous approaches are mainly to reduce the computational cost. Furthermore, as we treat the features separately at the spatial and channel levels, there is no spatial interaction involved throughout the whole procedure of channel shuffle. These differ from the previous use of DW Conv or channel shuffle.
>
> Although the proposed model seems simple, it requires a significant effort to optimize the model to achieve better results than SOTA methods. We sincerely hope the rebuttal would convince the reviewer to have a look at the paper as well as supplementary material and make a more convincing decision.
>
> **Q2: The authors argue that a larger receptive field is better for SR. While there are no experiments to analyze the difference between small and large kernels. For example, the authors can perform an ablation study by replacing the large kernels in the proposed network with small kernels and extending the depth to keep the model size.**
>
> As suggested, we perform experiments to explore the effect of increasing the depth (#B) of the proposed tiny model. The following experimental results show that this manner does not improve the reconstruction performance as clearly as using large kernels.
> | Model | #B | #C | Kernel Size | Params | FLOPs |DIV2K_val |
> |:-----|:-----:|:-----:| :-----:|:-----:|:-----:| :-----:|
> |Ours-Tiny | 5 | 32 | 3 | 113K | 8.9G | 28.78 |
> |A | 5 | 32 | 5 | 118K | 9.2G | 28.80 |
> |B | 5 | 32 | 7 | 141K | 10.4G | 28.85 |
> |C | 6 | 32 | 3 | 133K | 10.2G | 28.79 |
> |D | 8 | 32 | 3 | 174K | 12.8G | 28.81 |
>
> **Q3: How about the comparisons between speed?**
> As stated on L196 of the paper, the proposed models (tiny) need 0.016s to restore an HR image of $1280\times720$ pixels. As a comparison, the running times of CARN and LAPAR-A are 0.019s and 0.031s. This result indicates that the proposed model has a faster inference speed compared to the evaluated SOTA methods.
>
> **Q4: Large kernel size may heavily slow the whole network, which is also unfriendly to terminal devices.**
>
> Although using the convolution with large kernel size moderately increases the running time, benefiting from the proposed architecture, the running time of our method compares favorably against the competing methods as detailed above.

---

> > ### Author Response · Authors · 2022-08-07
> > **Follow-up to Reviewer YLsE**
> >
> > Thank you again for the constructive and thoughtful reviews. Please let us know whether the responses address your concerns. We will be happy to address any remaining concerns during the discussion.

---

### Official Review · Reviewer_xXRb · 2022-07-09

**Rating:** 7
**Confidence:** 5
**Soundness:** 4 excellent
**Presentation:** 4 excellent
**Contribution:** 3 good

**Summary:**

This paper proposes a ShuffleMixer architecture to combine large spatial depthwise conv and efficient channel and spatial mixing to perform efficient and state of the art single image super resolution results.

Edit: I think both the work in the paper and the authors' additional details and results allow me to update my overall paper rating.

**Questions:**

1. How do all of these results compare to bilinear/bicubic (Table 1 and Fig 1 results)? Having a non-learned baseline is an important comparison in fast SR results.

2. Reference [7] was an accepted paper to CVPR, please update the reference.

3. Does a different loss function increase the PNSR results? L1 is used both on pixel and fourier features, but would L2 on pixel help or hurt overall performance?

4. The paper boosts the performance of large kernels and reference [7] mentions 31x31 kernels. Why were ablations past 13x13 not performed?

5. Table 3 shows ablations for 3x3 to 13x13 kernels. Was a similar ablation done on the channel features for the model? E.g. is it more efficient to increase a model from 3x3 to 13x13 or to increase C appropriately to the same number of flops?

6. The introduction references mobile/embedded platforms with their high screen resolutions and the importance of super resolution. What is the runtime performance of these methods on such a mobile platform? Are there any practical considerations that large kernels or channel mixing have on devices with smaller caches?

7. For the results on Urban100, there is a note that the PSNR is lower than other methods. Is the PSNR low across the image, or are there patches where the PSNR-Y suffers more than others? I notice the patches chosen in the supplementary materials are very high frequency areas that I agree are perceptually similar to the state of the art method, but I wonder if a more in-depth analysis of the PSNR was done.

8. What is the importance of the SiLU vs other activation functions in this model? What is the performance gain over baseline ReLU? Are there efficiency concerns with implementing SiLU on mobile?

**Limitations:**

Authors sufficiently discuss limitations and impact of work.

**Strengths And Weaknesses:**

Strengths:
The Tiny model has strong results with a very small amount of compute and the regular sized model is competitive or is state of the with less compute.

This work introduces large receptive fields while keeping compute and parameters down while maintaining strong results.

Ablations on different parts of the architectures and kernel sizes show a principled search done in this architecture.


Weaknesses:
Poor PSNR/SSIM results on Urban100 (This is discussed in supplemental and I have follow up questions below).
No mobile runtime. (Importance of SISR in embedded / mobile discussed but no result shared. Additionally, line 78 declares a mobile friendly port).

---

> ### Author Response · Authors · 2022-08-02
> **Response to the Comments of Reviewer xXRb**
>
> We thank the reviewer for the positive comments on our work and answer the questions below.
>
> **Q1: How do all of these results compare to bilinear/bicubic (Table 1 & Fig 1)?**
>
> Our method performs better than the Bicubic one, as shown below. We will add the comparisons in the revised paper.
> | Method | Scale | Set5 | Set14 | B100 | Urban100 | Manga109 |
> |:-----|:-----:|:-----:| :-----:| :-----:|:-----:| :-----:|
> |Bicubic|$\times 2$|33.66|30.24|29.56|26.88|30.80|
> |Ours-Tiny|$\times 2$|37.85|33.33|31.99|31.22|38.25|
> |Bicubic|$\times 4$|28.42|26.00|25.96|23.14|24.89|
> |Ours-Tiny|$\times 4$|31.88|28.46|27.45|25.66|29.96|
>
> **Q2: Updating the reference [7].**
>
> Thanks, we will update it accordingly.
>
> **Q3: Does a different loss function increase the PNSR results? Would L2 on pixel help or hurt performance?**
>
> Different loss functions lead to different results. As the L1 loss is a commonly used one in image SR, we use it as the basic loss function in this work.
>
> We also evaluate the proposed method using the L2 loss on image pixels. The method using the L2 loss generates similar results to the method using the L1 loss (PSNR value: 28.787 vs. 28.776) on the DIV2K validation set.
>
> **Q4: Why were ablations past 13x13 not performed?**
>
> As suggested, we conduct experiments with larger kernel sizes ranging from 15 to 21. The below table shows that using a larger kernel size does not improve the performance significantly. This is mainly because ConvNets with larger kernels are difficult to train, as demonstrated in [7], and usually need some complex and sophisticated techniques to achieve desirable performance.
> | Kernel Size | Params | FLOPs | DIV2K_val |
> |:-----|:-----:|:-----:| :-----:|
> |$7 \times 7$|141K|10.7G|28.85|
> |$13 \times 13$|164K|12.2G|28.85|
> |$15 \times 15$|182K|13.4G|28.87|
> |$21 \times 21$|251K|17.9G|28.87|
>
> **Q5: Was a similar ablation done on the channel features for the model? E.g. is it more efficient to increase a model from 3x3 to 13x13 or to increase C appropriately to the same number of flops?**
>
> As suggested, we examine the effect of increasing the width (#C) of the proposed tiny model. The below table shows that increasing #C will rapidly increase the model parameters at the same model depth (#B) and the reconstruction performance are not as good as increasing the kernel size.
> | Model | #B | #C | Kernel Size | Params | FLOPs |DIV2K_val |
> |:-----|:-----:|:-----:| :-----:|:-----:|:-----:| :-----:|
> |Ours-Tiny|5|32|3|113K|8.9 |28.78|
> |A |5|32|5|118K|9.2G|28.80|
> |B|5|32|7|141K|10.4G|28.85|
> |C |5|36|3|138K|10.8G|28.81|
> |D |5|40|3|166K|13.0G|28.86|
>
> **Q6: What is the runtime performance of these methods on such a mobile platform?**
>
> As our work aims to design a lightweight and efficient SR model and most existing efficient SR methods are usually implemented based on NVIDIA GPUs, we follow the protocols of existing efficient SR methods and evaluate the inference time on an NVIDIA Tesla V100 GPU for fair comparisons.
>
> Moreover, converting the models that are implemented on NVIDIA GPUs into mobile platforms needs some specific operations, comparing the inference times of the methods without implementation on mobile platforms may not be fair. Thus, we do not report the inference time of these methods on the mobile platforms.
>
> **Q7: Are there any practical considerations that large kernels or channel mixing have on devices with smaller caches?**
>
> We do not specifically consider the practical application of large kernels on devices with smaller caches, as we mainly explore the feasibility of applying large kernel convolutions to lightweight SR network designs. Applying the algorithm to a specific device should be carefully optimized and tailored to the practical task.
>
> **Q8: [Results on Urban100] Is the PSNR low across the image or are there patches where the PSNR-Y suffers more than others? A more in-depth analysis of the PSNR is done.**
>
> We select several images with large PSNR differences (over 0.8dB) from the IMDN model for analysis. We crop selected images into patches and then calculate the corresponding PSNR values of these patches; the experimental results show that the PSNR is similar in most areas, but varies considerably in specific areas like edges.
>
> **Q9: What is the importance of SiLU? What is the performance gain over ReLU?**
>
> According to the definition of SiLU, it has the similar effect to a gating mechanism. Using SiLU in the proposed model (Ours-Tiny) leads to better results than ReLU (PSNR: 28.78 vs. 28.72).
>
> **Q10: Are there efficiency concerns with implementing SiLU on mobile?**
>
> As we aim to design a lightweight and efficient SR model and our method generates favorable performance in terms of model parameters, FLOPs, and inference time, we do not specifically consider the efficiency of SiLU on mobiles. Optimizing SiLU needs significant engineering efforts which is out of scope of the paper. This is a good question which is worth further investigation.

---

> > ### Author Response · Authors · 2022-08-07
> > **Follow-up to Reviewer xXRb**
> >
> > Thank you again for the constructive and thoughtful reviews. Please let us know whether the responses address your concerns. We will be happy to address any remaining concerns during the discussion.

---

### Official Review · Reviewer_X1ut · 2022-07-10

**Rating:** 5
**Confidence:** 4
**Soundness:** 2 fair
**Presentation:** 3 good
**Contribution:** 3 good

**Summary:**

The paper tackles one of the hot topics of computer vision, image super-resolution, which can be applied in different commercial usages (e.g. improving medical image analysis). The main characteristic of ShuffleMixer is the efficient construction of its architecture that uses depthwise convolution, projection layers, and mixing blocks.

**Questions:**

* Why did you evaluate Table 1 in Y-channel and Table 2 in RGB channels?
* What about table 3, is it in Y channel or RGB channel?


**Limitations:**

Yes

**Strengths And Weaknesses:**

Strengths:

* The manuscript is well structured and easy to follow  with its three proposals formed in a shuffle mixer layer, feature mixing block, and large kernel convolution.
* The proposed architecture only has 394 parameters (x2 scale) and is capable of running  on 91G FLOPs.
* The lightweight proposal, in x4 scale, achieves the state-of-the-art result, with less computational costs, in most of the standard datasets used for image super-resolution.

Weaknesses:

* Although the model is well structured in its proposal, some of its acronyms presented in the paper are not defined in the proper location. A deep revision of this lack is needed.
* Although the model is efficient and lightweight that outperforms the state of the art, since your proposal is the architectural settings, I feel that more experiments with different settings are needed. For example, the large kernel size goes till 13x13, why not evaluate, at least, till 15x15?. I see PSNR dropped when the 13x13  kernel size is considered but the explanation is not clear yet.
* ShuffleMixer is an important contribution. Therefore, a deep ablation study is needed, mixing the experiments with the settings of Table 2 and Table 3.

Minor errors:

* Abstract: what is the Fused-MBConv? Most people just read the abstract.
* As you stated in the first paragraph of your Introduction, it is well known the advantage of SR in Smartphones. However, a couple of citations are needed, or at least one, to show the formal usage in S22 Ultra / P50 Pro.
* 4 Experimental results: in this section experiments and their results are reported, please,  check the past time of some statements. For example, “We train our models on the DF2K dataset”.

---

> ### Author Response · Authors · 2022-08-02
> **Response to the Comments of Reviewer X1ut**
>
> We thank the reviewer for the comments on our work and answer the questions below.
>
> **Q1: Although the model is well structured in its proposal, some of its acronyms presented in the paper are not defined in the proper location. A deep revision of this lack is needed.**
>
> Thanks for the comments. We will carefully revise the paper to make it better understood.
>
> **Q2: Although the model is efficient and lightweight that outperforms the state of the art, since your proposal is the architectural settings, I feel that more experiments with different settings are needed. For example, the large kernel size goes till $13\times13$, why not evaluate, at least, till $15\times15$?. I see PSNR dropped when the $13\times13$ kernel size is considered but the explanation is not clear yet.**
>
> As suggested, we conduct experiments with different kernel sizes that range from $15\times15$ pixels to $21\times21$ pixels. The below table shows that using much larger kernel size does not improve the performance significantly. This is mainly because ConvNets with larger kernels are difficult to train as demonstrated [7] and usually need some complex and sophisticated techniques to achieve desirable performance. Given the model complexity and performance, we use the kernel size of $7\times7$ pixels for the proposed base model in the paper.
>
> | Kernel Size | Params | FLOPs | DIV2K_val |
> |:-----|:-----:|:-----:| :-----:|
> |$7 \times 7$|141K|10.7G|28.85|
> |$13 \times 13$|164K|12.2G|28.85|
> |$15 \times 15$|182K|13.4G|28.87|
> |$21 \times 21$|251K|17.9G|28.87|
>
> **Q3: ShuffleMixer is an important contribution. Therefore, a deep ablation study is needed, mixing the experiments with the settings of Tables 2 & 3.**
>
> In Tables 2 & 3 of the paper, we have conducted ablation studies to evaluate the effects of the main modules, i.e., the shuffle mixer layer, the feature mixing block, and a DW Conv with a large kernel.
>
> In addition, we further investigate the effect of increasing the depth (#B) or width (#C) of the proposed tiny model on the reconstruction performance. The below table shows that increasing the depth or width of the model with a small kernel does not yield as good a performance gain as enlarging the kernel size. These results also demonstrate the effectiveness of our proposed models. We will add corresponding discussions in the revised paper.
> | Model | #B | #C | Kernel Size | Params | FLOPs |DIV2K_val |
> |:-----|:-----:|:-----:| :-----:|:-----:|:-----:| :-----:|
> |Ours-Tiny | 5 | 32 | 3 | 113K | 8.9G | 28.78 |
> |A | 5 | 32 | 5 | 118K | 9.2G | 28.80 |
> |B | 5 | 32 | 7 | 141K | 10.4G | 28.85 |
> |C | 5 | 36 | 3 | 138K | 10.8G |28.81|
> |D | 5 | 40 | 3 | 166K | 13.0G | 28.86|
> |E | 6 | 32 | 3 | 133K | 10.2G | 28.79 |
> |F | 8 | 32 | 3 | 174K | 12.8G | 28.81 |
>
> **Q4: Why did you evaluate Table 1 in Y-channel and Table 2 in RGB channels? What about table 3, is it in Y channel or RGB channel?**
>
> Following the protocols of existing SR methods, e.g., the EDSR method, when performing quantitative comparisons between the proposed method and the evaluated ones, we calculate metric scores based on the Y channel. For the ablation studies of the proposed model (i.e., Tables 2 and 3), we use the RGB channel.
>
> We will add the results based on the Y channel to Tables 2 & 3 in the revised paper.

---

> > ### Author Response · Authors · 2022-08-07
> > **Follow-up to Reviewer X1ut**
> >
> > Thank you again for the constructive and thoughtful reviews. Please let us know whether the responses address your concerns. We will be happy to address any remaining concerns during the discussion.

---

> > > ### Comment · Reviewer_X1ut · 2022-08-10
> > > **Thank you for your clarification**
> > >
> > > Dear authors,
> > > thank you for your clarification, now I can see the strength of the proposal. However, I find more doubts. As you are presenting channel splitting,  shuffling strategy, and Fused-MBConv, how they can work with different kernel size? In the last table, the model D reaches 28.86, what is the behavior of PSNR if we change the kernel size to 7?
> > >
> > > Overall, it's a good contribution. Sorry for my late response, I was on vacation mode.

---

### Official Review · Reviewer_gjyf · 2022-07-13

**Rating:** 4
**Confidence:** 4
**Soundness:** 2 fair
**Presentation:** 3 good
**Contribution:** 2 fair

**Summary:**

This paper proposes an efficient image super-resolution network, namely ShuffleMixer. The core design of the network is the feature mixing block that learns a deep representation for the input images. The feature mixing block is inspired by recent architecture developments such as the layer normalization and MLP from transformers and the channel split and fusion from IMDN. To balance the parameter efficiency and the network performance, convolution with large kernels is used. It is reported that the proposed network achieves competitive to the state-of-the-art.

**Questions:**

1. The authors mentioned channel shuffling operation in the paper which is a submodule in the channel projection layer. As far as I understand, the channel shuffling operation is just a $1 \times 1$ convolution, which is quite different from the ShuffleNet operations. Thus, using the term of channel shuffle might be a little bit confusing.
2. Is there a discussion about the loss function? For example, how does the frequency constraint influence the performance of the network?
3. In the abstract, it is written that "Experimental results demonstrate that the proposed ShuffleMixer is about 6× smaller than the state-of-the-art methods in terms of model parameters and FLOPs while achieving competitive performance." Without mentioning the baseline network, this seems to be an overclaim.


**Ethics Review Area:**

["I don’t know"]

**Limitations:**

The authors did not discuss the limitation of their work. Considering the field of research, the discussion of potential negative societal impact is not necessary.

**Strengths And Weaknesses:**

Strengths:
1. The paper is well-written and easy to follow. The main figure clearly shows the design of the network.
2. Compared with the previous methods, the proposed method performs quite good under similar setting of number of parameters and computational complexity.

Weakness:
1. The paper is more about applications. Thus, it might fall out of scope for the conference that is more focused on theories.
2. Besides the number of parameters and computational complexity, the inference time is also factor that is at least of equal importance. The authors also mentioned the recent developments for efficient SR in AIM and NTIRE challenges. But unfortunately, there is no direct comparison with the top winners of the challenges (except IMDN) in Table 1 and Figure 1. In Figure 1, even IMDN is not listed, which is not preferrable. It is encouraged to evalutate different methods more thoroughly.
3. The maximum kernel size is $13 \times 13$. It not clear how the performance is if the kernel size is further enlarged.

---

> ### Author Response · Authors · 2022-08-02
> **Response to the Comments of Reviewer gjyf**
>
> We would like to thank the reviewer for taking the time to review our work and provide constructive comments. In the following, we address the concerns of the reviewers in detail.
>
>  **Q1: The paper might fall out of scope for the conference.**
>
> Although NeurIPS is a theory-oriented conference, it actually covers algorithms and applications. Moreover, there are lots of  papers on this topic that have been included by NeurIPS. Here we only show a small portion of the related papers in NeurIPS 2021.
>
> >[1] Zhang et al., Aligned Structured Sparsity Learning for Efficient Image Super-Resolution, NeurIPS, 2021.
>
> >[2] Yang et al., Implicit Transformer Network for Screen Content Image Continuous Super-Resolution, NeurIPS, 2021.
>
> >[3] Liu et al., Non-Local Recurrent Network for Image Restoration. NeurIPS, 2021.
>
> **Q2: Comparisons of the inference time.**
>
> As suggested, we compare the inference times of the proposed methods with those of the evaluated methods in the table below, and our methods achieve competitive performance.
> |Method|Scale|SR Size|Runtime(s)|
> |:-----|:-----:|:-----:|:-----:|
> |CARN |$\times 4$| $1280 \times 720$|0.019s|
> |LAPAR-A|$\times 4$|$1280 \times 720$|0.031s|
> |Ours-tiny|$\times 4$|$1280 \times 720$|0.016s|
>
> **Q3: Comparisons with the winners of the AIM/NTIRE challenge.**
>
> According to suggestions, we have compared with the winners in the AIM/NTIRE challenge. The following table shows that our method performs better than the winners in the previous AIM/NTIRE challenges.
> |Method|Scale|Params|FLOPs|Set5|Set14|B100|Urban100|Manga109|
> |:-----|:-----: |:-----:|:-----:|:-----|:-----:|:-----:|:-----:|:-----:|
> |PAN [4]|$\times 4$| 272K|28G|32.13|28.61|27.59|26.11|30.51|
> |RFDN [5]|$\times 4$| 550K|32G|32.24|28.59|27.54|26.15|30.48|
> |Ours-base|$\times 4$|411K| 28G| 32.21|28.66|27.61|26.08|30.65|
>
> >[4] Zhao et al., Efficient Image Super-Resolution Using Pixel Attention. ECCV Workshops, 2020.
>
> >[5] Liu et al., Residual feature distillation network for lightweight image super-resolution. ECCV Workshops,  2020.
>
> In the revised paper, we will add both quantitative and qualitative comparisons with the winners of the AIM/NTIRE challenges.
>
> **Q4: How is the performance if the kernel size is further enlarged?**
>
> We examine the effect of increasing the kernel size in the proposed method. The following table shows that using a larger kernel size does not improve the performance significantly. This is mainly because ConvNets with larger kernels are difficult to train as demonstrated in [7], that usually need some complex and sophisticated techniques to achieve desirable performance. Given the model complexity and performance, we use the kernel size of $7\times 7$ for the proposed base model in the paper.
>
> We will add the corresponding results in the revised paper.
>
> | Kernel Size | Params | FLOPs | DIV2K_val |
> |:-----|:-----:|:-----:| :-----:|
> |$7 \times 7$|141K|10.7G|28.85|
> |$13 \times 13$|164K|12.2G|28.85|
> |$15 \times 15$|182K|13.4G|28.87|
> |$21 \times 21$|251K|17.9G|28.87|
>
> **Q5: Using the term of channel shuffle might be a little bit confusing.**
>
> Similar to the shuffle operation in ShuffleNet, we use the channel shuffle for information exchange of grouped features. However, different from the ShuffleNet, the proposed method mixes channels on half of the split features by two $1\times1$ Conv layers and does not involve spatial information. Thus, we refer to it as “channel shuffle”. We will clarify this ambiguity in the revised paper.
>
> **Q6: Is there a discussion about the loss function? For example, how does the frequency constraint influence the performance of the network?**
>
> As stated on L157-158, we introduce a frequency domain loss to enhance high frequency components of images. Adding this loss function leads to a PSNR gain of about 0.1dB on the DIV2K validation set. We will add the corresponding discussion in the revised paper.
>
> **Q7: Without mentioning the baseline network in the final sentence of the abstract.**
>
> Our method is about 6 times smaller than LapSRN in terms of FLOPs for x4 SR and almost 3 times smaller than the classical lightweight methods, e.g., CARN, in terms of model parameters and FLOPs. We will fix the inaccurate expressions in the revised paper.

---

> > ### Author Response · Authors · 2022-08-07
> > **Follow-up to Reviewer gjyf**
> >
> > Thank you again for the constructive and thoughtful reviews. Please let us know whether the responses address your concerns. We will be happy to address any remaining concerns during the discussion.

---

### Author Response · Authors · 2022-08-09
**Follow-up to reviewers**

Thanks again to all reviewers for the constructive comments. The author-reviewer discussion is coming to an end, so please let us know if further clarification of previous concerns is needed.

---

### Meta-Review · Area_Chair_jRGP · 2022-08-27

**Recommendation:** Accept
**Confidence:** Certain

**Metareview:**

The paper received divergent reviews (with two reviewers leaning to reject and two leaning to accept). The key strengths of the paper are simple ideas and strong results on efficient and accurate image super-resolution.

The raised weakness include:
- "might fall out of scope for the conference" [Reviewer gjyf]
The AC disagrees with the statement. As authors' responses pointed out, CNN designs for computer vision applications are welcome at NeurIPS.

- "comparison with the top winners of the challenges" [Reviewer gjyf]
The authors' rebuttal include direct comparisons with previous winners.

- "how the performance is if the kernel size" [Reviewer gjyf]
The authors' provide the additional ablation.

There were no follow-up questions and no rebuttal acknowledgement from Reviewer gjyf. From the responses alone the AC thinks that the concerns have been sufficiently addressed.

Another major weakness is
- "The novelty of this paper may be limited. " [Reviewer YLsE].

It is true that basic components like depthwise convolution and channel shuffling have been explored in efficient CNN design. But as the reviewers' rebuttal described, the adoptation and modification of DW Conv or channel shuffle are non-trivial for the context of image SR.

In sum, the AC reads the reviews and rebuttal. Weighting both the strength and the weakness of the work, the AC decides to side with reviewer X1ut and xXRb and recommends to accept.


**Award:**

No

---

### Decision · Program_Chairs · 2022-09-14

Accept